# Diet Composition and Using Probiotics or Symbiotics Can Modify the Urinary and Faecal Nitrogen Ratio of Broiler Chicken’s Excreta and Also the Dynamics of In Vitro Ammonia Emission

**DOI:** 10.3390/ani13030332

**Published:** 2023-01-17

**Authors:** Nikoletta Such, Ákos Mezőlaki, Mohamed Ali Rawash, Kesete Goitom Tewelde, László Pál, László Wágner, Kornél Schermann, Judit Poór, Károly Dublecz

**Affiliations:** 1Institute of Physiology and Nutrition, Hungarian, Georgikon Campus, University of Agriculture and Life Sciences, Deák Ferenc Street 16, 8360 Keszthely, Hungary; 2Agrofeed Ltd., Duna Kapu Square 10, 9022 Gyor, Hungary; 3Regional Center for Food and Feed, Agricultural Research Center, Giza 12619, Egypt; 4Institute of Mathematics and Basics of Natural Sciences, Deák Ferenc Street 16, Georgikon Campus, 8360 Keszthely, Hungary

**Keywords:** ammonia emission, excreta composition, broiler chicken, probiotics, symbiotics, wheat, wheat bran

## Abstract

**Simple Summary:**

Ammonia emission is a main air quality issue worldwide and related mostly to animal production. This ratio in the European Union is about 80–90%. Ammonia is released from the manure, mainly from the nitrogen excreted via urine. There are several nutritional tools to decrease the N-excretion of the animals. Among them feeding low protein diets supplemented with crystalline ammino acids is probably the most efficient. However, feed additives, like exogenous enzymes, pro- and prebiotics can also modify the amount of the excreted N as well as its composition. There are lots of information on how the probiotics and prebiotics can modify the microbiota composition in the different gut segments. On the other hand, few results exist on how this modified microbiota can affect the ratio of the faecal and urinary nitrogen or the ureolytic activity of the excreted microflora. Therefore, in the present work the effects of different cereal grains with different soluble fibre fractions and a probiotic and symbiotic treatment was evaluated.

**Abstract:**

The objective of this research was to determine whether diet composition, or adding probiotic or symbiotic feed additives to broiler diets can modify the N composition of the excreta and the dynamics of ammonia volatilization from the manure. A total of 574 one-day-old Ross 308 broiler chickens were fed four different diets. The treatments included a corn and soybean meal-based control diets (C), wheat-based and wheat bran containing diets (W), a multi-strain probiotic treatment (Broilact^®^; Br), and a symbiotic additive containing *Bacillus subtilis*, inulin, and *Saccharomices cerevisiae* (Sy). Feeding the wheat-based diet significantly improved the weight gain and FCR of chickens. Treatment W also significantly increased the dry matter content of the excreta compared with the probiotic and symbiotic treatments. Both Br and Sy tended to decrease the amount of excreted uric acid, which is the main substrate of ammonia. Treatment Sy reduced the urinary N ratio of the excreta in comparison with treatment W. The symbiotic additive resulted in significantly higher ammonia emission in the first two hours. On the other hand, the dynamics of the emission was slow at the beginning and increased steeply after 15 h when the wheat-based diets were fed. Based on our results, the wheat-based diets, containing soluble arabinoxylans, and the symbiotic treatments of broiler diets have an impact on the urinary and faecal nitrogen composition of the excreta, and also on the dynamics of ammonia release from the manure.

## 1. Introduction

According to the National Emission Ceiling Directive 2016/2284, all member countries of the European Union must reduce their national emissions of air pollutants [1]. Among the member states, only a few countries have been able to reduce their emissions over the period of 2013–2018 [2]. Since agriculture and animal farming is responsible mostly for ammonia emissions, it is a very important task for researchers to develop practices that can be used to reduce ammonia emissions and to explore possible combinations of them. These practices should have their worth evaluated separately for each animal species due to digestive and anatomical differences. The European Union is one of the world’s largest poultry meat producers and a net exporter of poultry products with an annual production of around 13.4 million tons [3]. Ammonia emission is a concern for the poultry industry from both an environmental and an animal welfare point of view. Potential effects on the birds include respiratory diseases, viral infections, decreased production, and higher mortality [4]. The main source of ammonia production in broiler houses is the nitrogen found in urea, uric acid, and protein. These proteins are present in the undigested portion of excreta and are derived from amino acid-rich diets that ensure that the nutritional needs of the birds are met [5,6]. Based on the available research results, uric acid represents 50–60% of the total N content of poultry excreta [7]. With the development of nutrition and genetics, this ratio may have changed, but only a few results are available concerning this topic in the literature. According to studies some dietary treatments can modify this ratio [8]. Such feed supplements include pre- and probiotics, which have been introduced to replace antibiotics [9,10,11]. However, a few studies have shown that in addition to improving intestinal health, they can also affect faecal ammonia emissions. The results of studies evaluating biological agents to reduce ammonia emissions from poultry litter are contradictory and the reason for the reduction in ammonia emissions is not clear [12]. There are also manure treatment technologies that can be used efficiently. For example, adding zeolite or bentonite to the manure can decrease the ammonia release in the poultry house [13], The ammonia production and gas emissions are caused through the action of microorganisms having urease and deaminase activities [14]. A possible way to alter the intestinal flora is to use prebiotic feeds. Maize is the preferred grain for feeding poultry because its dietary energy value is the highest among cereals with very low variability between years for a given region [15]. In certain regions, wheat is used because of availability or artificial price support and artificial import tariffs on alternative ingredients. There is renewed interest in fibre nutrition of all classes of poultry, in terms of both gut health and effect on microflora [16]. Beside the well-known prebiotics, the soluble arabinoxylan of wheat also has a prebiotic effect if the diets are supplemented with exogenous xylanase [17]. Oligosaccharides formed because of fibre degradation and are primary substrates for the growth of intestinal microorganisms [18]. Fermentable oligosaccharides increase the concentration of short chain fatty acids (SCFA) in the cecum and colon, which decrease the pH and allow for the colonization of the potential harmful bacteria [19].

Another option is to use probiotics. The impact on the urinary and faecal N excretion in chickens has not been completely clarified yet. Probiotic bacteria can modify protein digestibility and the proteolytic breakdown of the non-digested protein in the hind gut segments [20]. Previous studies have reported that the *Bacillus subtilis* reduced ammonia emission in broilers as a direct-fed microbial or probiotic product [21,22]. A previous experiment was run to study the possibility of elimination of uric acid from poultry manure by using it as a medium for single cell protein (SCP, yeast) production. It was found that yeasts make efficient use of uric acid from poultry manure, thereby eliminating its environmental pollution [23]. However, according to the literature, the use of multi-strain probiotics is more efficient than using mono-strain probiotics, because different strains of the genus show symbiotic or additive relationships towards each other, which positively affects the microbial community [24]. Similarly, many authors agree that a symbiotic product consisting of a combination of synergistically interacting probiotics and prebiotics may provide a better efficacy in the stimulation of intestinal microbiota and protection of animal health compared to the separate application of probiotics and prebiotics [25,26,27]. In contrast to the continuous feeding of pro- and/or prebiotics during the whole fattening period, a single inoculation of microbes at an early age of birds is another possible way to stimulate the growth and activity of beneficial microflora in the digestive tract. The competitive exclusion (CE) products composed of stable, mixed microbes derived from the intestinal microbiota of healthy adult animals and their application is based on the so-called Nurmi concept [28]. The inoculation of CE products in ovo or directly upon hatching may be a viable method to aid in the early development of a microbial population [29,30,31], which may affect the release of ammonia from the faeces.

The aim of this study is to examine the effect of a prebiotic feed component (wheat and wheat bran), a CE culture, and a symbiotic treatment (*Bacillus subtilis*, inulin, *Saccharomyces cerevisiae*) on production traits, specifically the amount of excreted nitrogen forms, the ratio of faecal and urinary nitrogen, and the dynamics of ammonia emission from the excreta of broiler chickens.

## 2. Materials and Methods

### 2.1. Birds and Experimental Design

A floor pen trial was carried out at the experimental farm of the Institute of Physiology and Nutrition, Hungarian University of Agriculture and Life Sciences (Georgikon Campus, Keszthely, Hungary). The animal experiment was approved by the Institutional Ethics Committee (Animal Welfare Committee, Georgikon Campus, Hungarian University of Agriculture and Life Sciences). A total of 574 one-day-old Ross 308 broiler chickens were purchased from a commercial hatchery (Gallus Ltd. Devecser, Hungary) and placed into 24 floor pens, 24 chickens per pen (10 chickens/m^2^). Each treatment was replicated 6 times. Chopped wheat straw was used as litter material.

A maize–soybean-based basal diet was fed without feed additive to the control group (C). Birds of the second treatment (Br) were fed the control diet, and a solution of the product Broilact^®^ was given to the birds via crop inoculation in two equal doses (1.25 × 10^7^ CFU/0.5 mL) at day 0 and 1. All the chickens of the two other treatment groups were inoculated with drinking water. The product Broilact^®^ (Europharmavet Ltd., H-1077 Budapest, Rózsa str. 10–12., Hungary) is a refined gut microbiota derived from healthy adult hens and was screened to ensure the absence of specific pathogens [32]. The basal diet was supplemented with a symbiotic additive mixture in the third treatment group (Sy) and fed throughout the whole trial. The symbiotic additive mixture contained three products: GalliPro^®^200, at a dose of 0.4 g/kg diet (Bacillus subtilis, DSM17299 bacterial strain; 1.6 × 10^6^ CFU/g, Biochem Ltd., Küstermeyerstrasse 16. 49393 Lohne, Germany); Orafti^®^ HSI containing inulin, at a dose of 5 g/kg diet (Beneo Ltd., Aandorenstraat 1, B. 3300 Tienen, Belgium); and a yeast-product, Levucell^®^ SB20, at a dose of 0.05 g/kg diet, providing 1 × 10^9^ CFU viable yeast cells per kg of diet (Saccharomyces cerevisiae boulardii, 2 × 1010 CFU/g, Lallemand Ltd., Ottakringer Str. 89, A-1160 Vienna, Austria). Birds of the fourth treatment (W) were fed a wheat-based diet, which contained 30% wheat; the wheat bran content of the starter diets was 3% and those of the grower and finisher diets were 6%, respectively.

Three phases of fattening were used. The starter diets (0–10 days) were fed in mash; the grower (11–24 days) and finisher feeds (25–40 days) were in pelleted form. Cold pelleting was used without hydrothermal pre-treatment, and the temperature of the pellets were below 60 °C. Feed and water were available ad libitum. Diets were formulated to be isoenergetic and isonitrogenous, and the nutrient content of the diets met the requirements of Ross 308 broiler chickens. The composition and nutrient content of experimental diets is shown in Table 1 and Table 2. Computer-controlled housing and climatic conditions were maintained during the trial according to the breeder’s recommendations [33]. The room temperature was adjusted to 34 °C at day 1 and reduced gradually to 24 °C at 18 days of age. Each pen was lined with chopped straw as bedding material. Pens were equipped with one tube feeder and one cup drinker until 10 days of age and bell drinkers were used onward. The light intensity was 30 lux in the first week, and 10 lux thereafter with constant day length of 23 h from 0–7 days and 20 h light and 4 h dark period thereafter. Both feed and water were available ad libitum throughout the experimental period.

### 2.2. Measurements

During the 40-day-long fattening period, the body weight (BW) of all animals was measured at the beginning (day 0) and at the end (day 40) of the trial. Feed intake (FI), body weight gain (BWG), feed conversion ratio (FCR), and mortality were calculated on pen basis. On day 40, about 200 g fresh excreta samples were collected. Clean nylon foils were placed on the half surface area of the pens and 8 birds were separated to this area. Under constant supervision, the excreta were collected immediately after defaecation from at least 5 birds per pen. The samples of the 5 birds were pooled, mixed thoroughly, frozen, and stored at −20 °C until further processing. From these samples, their dry matter, total N, NH_4_^+^-N, and uric acid-N contents were measured. The dry matter content of excreta samples was measured in exicator (100 °C for 24 h). The urinary N was calculated as the sum of uric acid-N and NH^4+^-N as described by O’Dell et al. [34]. Total N was determined according to the Kjeldahl method with a Foss-Kjeltec 8400 Analyzer Unit (Nils Foss Allé 1, DK-3400 Hilleroed, Denmark). The determination of NH_4_^+^-N content of the excreta was carried out according to the method of Peters [35]. The uric acid measurement based on the method of Marquardt et al. [36]. All N parameters were adjusted to dry matter basis. The sum of ammonium-N and uric acid-N was considered as urinary N content [34]. The in vitro ammonia emission measurement was carried out at five time points using the method of Santoso [37]. The ammonia concentration of the air was measured with a Draeger equipment (model X-am 5600; Drägerwerk AG & Co. KGaA, Lübeck, Germany) with a compatible Dräger X-am^®^ external pump. The pump was designed for clearance measurements, for example, taking air samples form a container or tank. Samples were thawed and 50 g homogenized excreta samples were placed into 1 L double-sealed plastic containers. Each container was equipped with a cover containing a hole to allow insertion of a gas measuring tube that was sealed inside with adhesive plaster. Measurements were taken five times: 1, 2, 4, 15, and 17 h after entering the tank. The ammonia measurement range of the equipment’s sensor was 0–300 ppm. The adhesive plaster was punctured, and 1000 mL of headspace air was collected from approximately 10 cm above the sample surface. After sampling, the tubes were sealed again. The concentration of NH_3_ was expressed as milligrams per litre.

### 2.3. Statistical Analysis

The experimental unit in all the measured parameters was the pen. All data were analysed by ANOVA, using the SPSS 22.0 software (IBM Corp. 1 New Orchard Road, Armonk, New York 10504-1722, USA). Differences were considered as significant at a level of *p* ≤ 0.05. Differences at *p* ≤ 0.1 were defined as a tendency.

## 3. Results

Feeding of the wheat-based and wheat bran supplemented diets significantly improved the final body weight, the weight gain, and the feed conversion ratio of the chickens, compared to the control, the Broilact^®,^ and the symbiotic supplemented diets. There was no difference in the feed consumption of birds (Table 3). The mortality of all treatment groups was low; in the C, W, Br, and Sy groups only 1, 2, 3 and 1 chicken died, respectively.

Dietary treatments failed to modify the total N and NH_4_^+^-N of excreta (Table 4). On the other hand, as a tendency, the uric acid-N content of excreta samples was lower in the probiotic and symbiotic treatments. The dry matter content of the excreta samples was only affected by wheat. This treatment resulted in significantly higher dry matter content compared with the Br and Sy treatments.

Since treatments Br and Sy reduced the amount of uric acid-N, the percentage of the urinary-N was also the lowest for these two treatments (Figure 1). The symbiotic treatment had the lowest urinary N-ratio, which was significantly lower than that of the W treatment. A similar trend was observed with the Broilact^®^ treatment, but in this case the difference was not significant.

The in vitro ammonia emission data are shown in Table 5. After one hour incubation, the ammonia release in the case of Sy was significantly higher than those of the C and W treatments. After 2 and 4 h this difference remained significant only between the Sy and W treatments. Thereafter, at 15 and 17 h, the differences were not statistically significant. The dynamics of ammonia emission show that the emission of treatment Sy was steadily higher than those of treatment C and treatment Br. On the other hand, the ammonia release from the excreta of the wheat-based diet was slow in the first hours, but increased steeply after 4 h and resulted the highest emission after 17 h.

## 4. Discussion

Using pro- and prebiotics or symbiotics as antibiotic alternatives is a common practice in poultry nutrition. There is plenty of information on the effects of these feed additives on the gut microbiota composition and production parameters of broiler chickens [25,26,27]. However, little information is available on how the changes in gut microflora can affect the nitrogen metabolism, the excreted amount, the ratio of the faecal and urinary N, and the manner the ammonia emission from the chicken manure.

Broilact^®^ failed in this trial to modify the production traits, which is in agreement with most of the published data [38,39]. Symbiotics can be effective both in the small intestine and in caeca and in some cases improve also the growth or the feed conversion of birds. However, the results are contradictory because of the composition and concentration of the ingredients, the age of the birds, the diet composition, and the method of treatments [37]. In our case the symbiotic treatment did not modify the production parameters compared with the control or the Broilact^®^ supplemented diets.

Wheat-based diets, because of their soluble NSP content, can decrease the digestibility of nutrients and impair the production traits of animals [40]. Using exogenous xylanase, however, can dissolve this problem. Xylanase in wheat-based diets not only decrease viscosity, but also provide such xylan-oligosaccharides (XOS), that can play as prebiotics in the hind gut [41].

In this trial the wheat-based diet with wheat bran supplementation improved the weight gain and feed conversion of broiler chickens significantly compared with a commercial corn and soybean meal-based diet. Since the nutrient content of the experimental diets was almost identical, the positive effects in the production traits could be explained at least partly by the prebiotic effects of XOS when wheat-based diets with xylanase supplementation were used. XOS can modify the microflora and thus the volatile fatty acid production in the caeca [17]. Furthermore, in the small intestine the reduction of viscosity improves digestion as well as weight gain and feed conversion [42].

Wheat contains soluble arabinoxylans and can increase digesta viscosity and the water content of the excreta; in the present study, treatment W increased the dry matter of the chicken’s excreta compared with the other treatments. Higher dry matter content improves litter quality and mitigates ammonia emission from the manure [43]. It can also decrease the incidence of foot pad lesions [44]. The dry matter content of excreta however, is influenced not only by the faeces but also by the urine content. This parameter is difficult to measure in poultry because of the cloaca. However, since feeding wheat increases the fermentation in the caeca and more ammonia is probably converted to bacterial protein, less uric acid is synthetized in the liver and this way less urine is excreted. In our previous study, a corn-based diet supplemented with wheat bran did not modify the dry matter content of the excreta. However, in that treatment no wheat was used, so the soluble fibre and arabinoxylan contents were less. In that trial, feeding a butyric acid-producing bacteria increased the excreta’s dry matter [8]. Grashorn et al. [43] reported that the dry matter content of the chymus was significantly higher in all segments of the digestive tract in the experimental group with acidified feed than in the control group. Since wheat and its soluble XOS can increase the butyrate production of the caeca [41], our result could also be explained accordingly, but the exact interaction between butyrate and the amount of excreted water remains unclear.

Any change in gut microflora affects microbial composition of faeces and the quantity of gases released from the manure [45]. We have found in our previous study that wheat bran supplementation of maize-based diets increases the urease activity of excreta, affecting the amount and dynamics of ammonia emission [8]. The result of this study partly confirms that feeding wheat-based or wheat bran supplemented diets with xylanase increases the ammonia production from the faeces. Although in this study wheat-based diets initially had the lowest ammonia release, after time passed it became the highest. The initial slower production of ammonia may be due to the higher dry matter content of the excreta in the wheat-based treatment.

*Bacillus subtilis* bacterial strain was used in our symbiotic treatment. It is one of the three bacteria that showed the best inhibitory effect on NH_3_ formation in an experiment by Mi et al. [46]. Santoso et al. [37] observed that feeding a dried culture of *Bacillus subtilis* significantly reduced NH_3_ gas release, perhaps through suppression of the urease-producing microflora. In addition, supplementation of duck diets with *Bacillus subtilis* reduced serum ammonia and blood uric acid levels [47]. Other experiments have shown that the other components of symbiotic treatment, inulin, and yeast can also reduce NH_3_ production [48,49]. Yeasts of the genus *Saccharomyces* are known to synthesize amino acids using organic and inorganic nitrogen sources [13,50]. The *Saccharomyces* species may also have potential efficacy in the inhibition of harmful ammonia-producing bacteria and pathogens [13]. *Saccharomyces* spp. strains are facultative anaerobic microbes capable of proliferating under both aerobic and anaerobic conditions [51]. It was also confirmed that *Saccharomyces cerevisiae* can grow also in the manure under all tested conditions [13]. In an experiment with dairy cows, the yeast culture (*Saccharomyces cerevisiae*) tended to reduce rumen ammonia concentration and increased microbial protein synthesis, and decreased ammonia and methane emissions from manure [52]. In our experiment, the ammonia emission-reducing effect of *Bacillus subtilis* was not observed; moreover, the release of ammonia was fastest with this treatment. Nitrogen-containing compounds from animal production are converted to gaseous ammonia by microbial activity [53,54]. Much of the ammonia released from manure comes from the hydrolysis of urea [55], or in the case of birds from the breakdown of uric acid [37]. Based on the available research results, uric acid represents 50–60% of the total N content of poultry excreta [44]. O’Dell et al. [34] found that the sum of uric acid and NH_4_^+^-N of the excreta gives approximately the total amount of urinary nitrogen in birds. Our results suggest that the components of the symbiotic treatment increased the conversion of ammonia to bacterial protein in the cecum. This may have led to a decreased urinary N ratio. In the case of the Broilact^®^ treatment, the increase was not significant. This result is consistent with the results obtained in our previous trial when the lactic acid-producing bacteria *(Lactobacillus farciminis* CNMA67-4R) treatment increased the proportion of faecal nitrogen in the excreta [8].

## 5. Conclusions

From the results of this trial we can conclude, that pro- and prebiotics or symbiotics have only marginal effects on the production traits, but feeding wheat and wheat bran containing diets with xylanase supplementation can improve the growth rate and feed conversion of broiler chickens. The cereal grain composition of diets in this trial did not modify the faecal and urinary N ratio of the chicken’s excreta. On the other hand, the symbiotic treatment significantly decreased the uric acid content and the urinary N ratio of the excreta, which is a positive change from an ammonia emission point of view. Therefore, diet composition and using different feed additives can not only modify gut health and the production parameters, but also the caecal and excreted microbiota and the dynamics of ammonia emission from the manure.

## Figures and Tables

**Figure 1 animals-13-00332-f001:**
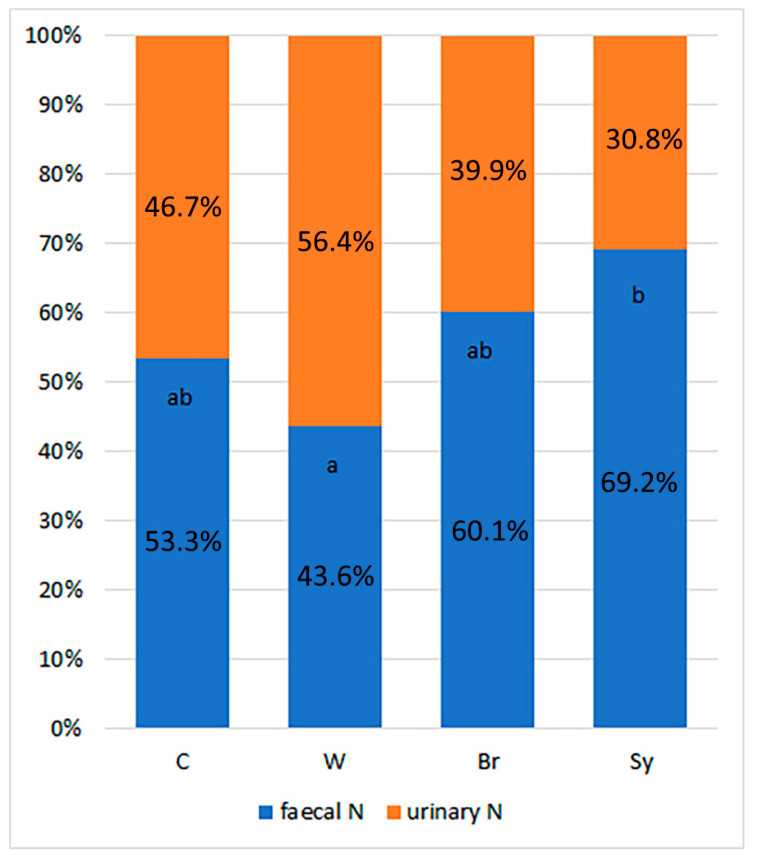
Effects of dietary treatments on the faecal and urinary nitrogen ratio of excreta. C—control diet, W—wheat-based diet, Br—control diet with Broilact^®^ supplementation, Sy—control diet with symbiotic supplementation. SEM—standard error of the mean; ^a,b^ means with different superscripts are significantly different (*p* < 0.05).

**Table 1 animals-13-00332-t001:** Composition of experimental diets (g/kg as fed).

	Starter (Day 1–10)			Grower (Day 11–24)			Finisher (Day 25–40)		
	C/Br	W	Sy	C/Br	W	Sy	C/Br	W	Sy
Corn	431	140	425	488	166	480	547	229	540
Wheat	-	300	-	-	300	-	-	300	-
Wheat bran	-	30	-	-	60	-	-	60	-
Extracted soybean meal	464	410	464	412	353	412	358	307	358
Sunflower oil	56	68	56	58	77	58	55	65	55
Limestone	18	18	18	15	15	15	14	14	14
MCP	16	16	16	15	14	15	14	13	14
L-lysine	2	3	2	1	2	1	1	2	1
DL-methionine	4	4	4	3	3	3	3	2	3
L-threonine	-	1	-	1	1	1	0.5	0.5	0.5
L-valine	-	1	-	-	0.5	-	-	0.5	-
NaCl	3	3	3	3	3	3	3	3	3
NaHCO_3_	1	1	1	1	1	1	1	1	1
Premix ^1^	5	5	5	5	5	5	5	5	5
Phytase ^2^	0.1	0.1	0.1	0.1	0.1	0.1	0.1	0.1	0.1
NSP enzyme ^3^	0.1	0.1	0.1	0.1	0.1	0.1	0.1	0.1	0.1
Probiotic ^4^	-	-	0.4	-	-	0.4	-	-	0.4
Inulin ^5^	-	-	5	-	-	5	-	-	5
Yeast ^6^	-	-	0.05	-	-	0.05	-	-	0.05
Sum	1000	1000	1000	1000	1000	1000	1000	1000	1000

(C) control diet, (Br) control supplemented with Broilact^®^, (W) wheat-based diet supplemented with wheat bran, (Sy) control diet supplemented with symbiotic; ^1^ Premix was supplied by UBM Ltd. (Pilisvörösvár, Hungary). The active ingredients in the premix were as follows (per kg of diet): retinyl acetate—5.0 mg, cholecalciferol—130 μg, dl-alpha-tocopherol-acetate—91 mg, menadione—2.2 mg, thiamine—4.5 mg, riboflavin—10.5 mg, pyridoxin HCL—7.5 mg, cyanocobalamin—80 μg, niacin—41.5 mg, pantothenic acid—15 mg, folic acid—1.3 mg, biotin—150 μg, betaine—670 mg, Ronozyme^®^ NP—150 mg (DSM Nutritional Products, 1416 North Williamsburg County Highway Kingstree SC 29556, United States), monensin-Na—110 mg (only grower), narasin—50 mg (only starter), nicarbazin—50 mg (only starter), antioxidant—25 mg, Zn (as ZnSO_4_·H_2_O)—125 mg, Cu (as CuSO_4_·5H_2_O)—20 mg, Fe (as FeSO_4_·H_2_O)—75 mg, Mn (as MnO)—125 mg, I (as KI)—1.35 mg, Se (as Na_2_SeO_3_)—270 μg; ^2^ Quantum Blue^®^ 5G (AB Vista, Marlborough, Wiltshire, SN8 4AN, England); ^3^ Econase^®^ XT 25P (AB Vista, Marlborough, Wiltshire, SN8 4AN, England); ^4^ GalliPro^®^ 200 (Bacillus subtilis, DSM17299 bacterial strain; 1.6 × 10^6^ CFU/g, Biochem Ltd., Küstermeyerstrasse 16. 49393 Lohne, Germany); ^5^ Orafti^®^ HSI (Beneo Ltd., Aandorenstraat 1, B. 3300 Tienen, Belgium); ^6^ Levucell^®^ SB20 (Saccharomyces cerevisiae boulardii, 2 × 10^10^ CFU/g, Lallemand Ltd., Ottakringer Str. 89, A-1160 Vienna, Austria).

**Table 2 animals-13-00332-t002:** Calculated and measured nutrient contents of the experimental diets (g/kg as fed).

	Starter (Day 1–10)			Grower (Day 11–24)			Finisher (Day 25–40)		
	C/Br	W	Sy	C/Br	W	Sy	C/Br	W	Sy
Calculated nutrient content									
AMEn (MJ/kg)	12.65	12.65	12.65	13.0	13.0	13.0	13.2	13.2	13.2
Crude protein	230	230	230	210	210	210	190	190	190
Crude fibre	28.52	30.28	28.52	27.54	31.32	27.54	26.51	27.02	26.51
Ca	10.5	10.5	10.5	9.0	9.0	9.0	8.5	8.5	8.5
non-phytate P	5.0	5.0	5.0	4.5	4.5	4.5	4.2	4.2	4.2
Lys	14.11	14.10	14.11	12.26	12.28	12.26	10.82	10.79	10.82
Met	6.83	6.69	6.83	6.00	5.86	6.00	5.37	5.19	5.37
M+C	10.39	10.37	10.39	9.31	9.31	9.31	8.41	8.38	8.41
Thr	9.73	9.66	9.73	8.60	8.55	8.60	7.48	7.41	7.48
Val	10.83	11.18	10.83	9.94	9.98	9.94	9.01	8.91	9.01
SID Lys	12.70	12.70	12.70	11.00	11.00	11.00	9.70	9.70	9.70
SID Met	6.54	6.38	6.54	5.73	5.56	5.73	5.12	4.92	5.12
SID M+C	9.40	9.40	9.40	8.40	8.40	8.40	7.60	7.60	7.60
SID Thr	8.30	8.30	8.30	7.30	7.30	7.30	6.32	6.32	6.32
SID Val	9.53	9.50	9.53	8.77	8.40	8.77	7.97	7.50	7.97
Measured nutrient content									
Dry matter	903	905	903	898	902	898	897	897	897
Crude protein	243	239	242	223	220	220	197	206	200
Crude fat	72	83	75	77	91	78	72	63	84
Crude fibre	38.0	42.3	39.3	31.6	38.9	32.7	33.5	32.8	38.2
Ash	69.6	70.0	69.0	63.9	63.2	64.2	57.5	58.8	58.7
Ca	10.4	10.7	10.4	9.5	9.7	9.5	9.0	9.7	8.2
P	6.9	7.6	7.1	6.8	7.1	6.7	6.7	6.7	7.1
Starch	314	308	313	347	323	334	381	372	363
Lys	14.1	13.8	14.0	12.7	12.5	12.6	11.0	11.6	11.3
Met	6.7	6.6	6.8	6.0	6.0	5.9	5.6	5.6	4.8
Cys	3.7	3.8	3.8	3.4	3.5	3.4	3.1	3.2	3.3
Thr	10.0	9.8	10.4	9.3	9.0	9.3	7.9	8.3	7.8
Val	11.1	11.6	11.0	10.2	10.5	10.2	9.1	9.5	9.7

**Table 3 animals-13-00332-t003:** The effect of dietary treatments on the production parameters of broiler chickens.

Treatments	Body Weight at Day 40 (g)	Body Weight Gain (g/bird)	Cumulative Feed Intake (g)	FCR (kg/kg)
	g/birds			kg/kg
C	2397 ^b^	2354 ^b^	4020	1.59 ^a^
W	2553 ^a^	2509 ^a^	3913	1.43 ^b^
Br	2398 ^b^	2355 ^b^	4038	1.58 ^a^
Sy	2441 ^b^	2398 ^b^	4026	1.55 ^a^
SEM	18.57	18.57	42.51	0.10
*p*-values	0.002	0.002	0.734	0.026

C—control diet; W—wheat-based diet; Br—control diet with Broilact^®^ supplementation; Sy—control diet with symbiotic supplementation; FCR—feed conversion ratio (kg feed/kg live weight); SEM—standard error of the mean; ^a, b^ means with different superscripts are significantly different (*p* < 0.05).

**Table 4 animals-13-00332-t004:** The effect of dietary treatments on the N-forms and dry matter content of excreta.

Treatment	Total N	NH_4_^+^-N	Uric Acid-N	Dry Matter
	mg/g Dry Matter			%
C	45.42	3.81	17.53 ^A^	15.17 ^ab^
W	41.23	4.54	17.6 ^A^	16.16 ^a^
Br	48.36	3.73	14.60 ^B^	14.72 ^b^
Sy	50.21	2.58	12.62 ^B^	14.81 ^b^
Pooled SEM	1.82	0.42	0.83	0.18
*p*-values	0.341	0.468	0.090	0.015

C—control diet; Br—control diet with Broilact^®^ supplementation; W—wheat-based diet; Sy—control diet with symbiotic supplementation; SEM—standard error of the mean; ^a, b^ means with different superscripts are significantly different (*p* < 0.05); ^A, B^ means trends (*p* < 0.1).

**Table 5 animals-13-00332-t005:** Treatment effects on the in vitro NH_3_ emission.

Treatment	1 h	2 h	4 h	15 h	17 h
mg/L NH_3_					
C	8.8 ^b^	14.3 ^ab^	26.0 ^ab^	48.0	50.6
W	7.0 ^b^	13.0 ^b^	20.4 ^b^	63.0	65.8
Br	9.2 ^ab^	16.8 ^ab^	25.4 ^ab^	51.1	50.3
Sy	13.8 ^a^	23.8 ^a^	36.3 ^a^	53.1	52.8
Pooled SEM	0.80	4.05	2.17	3.79	4.24
*p*-values	0.006	0.033	0.049	0.562	0.549

C—control diet, W—wheat-based diet, Br—control diet with Broilact^®^ supplementation, Sy—control diet with symbiotic supplementation. SEM—standard error of the mean; ^a, b^ means with different superscripts are significantly different (*p* < 0.05).

## Data Availability

All data generated or analysed during this study are included in this published article.

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
