# Peer review of "Diet Composition and Using Probiotics or Symbiotics Can Modify the Urinary and Faecal Nitrogen Ratio of Broiler Chicken’s Excreta and Also the Dynamics of In Vitro Ammonia Emission"

_animals, 2023, doi:10.3390/ani13030332_

Round 1
Reviewer 1 Report
This study aims to evaluate the effect of diet composition and probiotics or symbiotics on the urinary and faecal nitrogen ratio of broiler chicken’s excreta and also the dynamics of in vitro ammonia emission. Although the subject is interesting the data presented is very limited and does not allow solid conclusions.
My main concern is sample collection. It was exclusively done at the end of the experiment in a single day. At least a three-day collection should be desirable to guaranty the results are representative.
No data of mortality and productive parameters of the animals during the experiment is given. Performance data would be very useful to make a better discussion of the results.
The wheat based diet it is indicated to have a higher content of soluble fiber but no analysis have been provided
The discussion is very poor and in many aspects completely speculative. No conclusions are given.
Overall, this paper, as it is presented now, does not significantly contribute to the advance of knowledge
Reviewer 3 Report
The authors conducted interesting research. Unfortunately, there are some errors that need to be corrected. The experimental group as positive and negative control is missing. all noted in the publication.

Round 2
Reviewer 2 Report
It is acceptable now, with minor changes
Table 3.... At the top: Feed intake Cum. instead of Feed intake cumulative. Variable; feed convertion ratio (FCR) should be described below with the unit, kg of feed/kg of meat??
The Reference section has format problems, some references (32-42) moved to the left.
Reviewer 3 Report
The authors introduced a lot of corrections, which improved the readability of the publication. However, no explanation was obtained on what basis the measurement of ammonia in 1000 ml of air is estimated, how it was determined that it is such a value. Is this based on the manufacturer's data or did the authors install a flowmeter? It also proposes to update items 12 and additionally introduce the following more recent publication from 2016: Łukasz Wlazło et al . Removal of ammonia from poultry manure by aluminosilicates. J. Environ. Manag. 2016,Vol 183 , 722-725, DOI: 10.1016/j.jenvman.2016.09.028
Round 3
Reviewer 3 Report
thank you for all the explanations,